# Macroeconomic factors affecting FDI in the African region

**Sashini Rathnayake**[1], **Sanjula Jayakody**[1], **Pasindu Wannisinghe**[1],
**Deshani Wijayasinghe**[1], **Ruwan Jayathilaka**[2]*, **Naduni Madhavika**[2]

**1** SLIIT Business School, Sri Lanka Institute of Information Technology, Malabe, Sri Lanka, **2** Department of Information Management, SLIIT Business School, Sri Lanka Institute of Information Technology, Malabe, Sri Lanka

* ruwan.j@sliit.lk

**Data Availability Statement:** All relevant data are within the paper and its with Supporting Information files.

**Funding:** The authors received no specific funding for this work.

## Abstract

Foreign Direct Investment (FDI) occurs when one country invests in another. Multiple factors have contributed to fluctuations in FDI flows globally. This study investigates the impact of the Logistics Performance Index (LPI), Global Competitiveness Index (GCI) and Interest Rates (IR) on FDI in the African region. The study is significant because the African region is underdeveloped and with an unstable macroeconomic environment. Data were collected for 26 countries in the African region for the years 2007, 2010, 2012, 2014, 2016 and 2018 and analysed using Panel Regression and Multiple Linear Regression models. The study's findings concluded that LPI, GCI, and IR are three major macroeconomic factors impacting FDI inflows. The results indicated that LPI positively impacts FDI in Gambia, Lesotho and Rwanda, while in contrast, LPI impacts FDI negatively in Mauritius. GCI has a positive impact on FDI in Algeria and Lesotho with a negative impact in Rwanda, Mauritius and Namibia. Moreover, IR has a negative impact on FDI in Algeria, Rwanda and Mauritius with a positive impact in Lesotho. Policymakers should pay more attention to the infrastructure development and management of macroeconomic and other factors affecting FDI.

## Introduction

The African region is home to underdeveloped countries afflicted by terrorism, political unrest, and extreme poverty. Despite its abundant natural resources and strategic geographical location, African region economies have been fragile, indicating poor macroeconomic conditions which have deprived the quality of life of their citizens. As a result, the African region ranks low in all indexes and only attracts a negligible amount of Foreign Direct Investment (FDI). Therefore, it is necessary to identify how the global indicators and Interest Rates (IR) impact the FDI flows in the African region. FDI is a type of investment comprised of global capital flows and is crucial to region's economic growth. According to Dhahri and Omri [1], FDI is more than just a transfer of capital worldwide; it is also a type of global production. According to existing studies, there is a connection between FDI and the economy. A country's economic development is primarily reliant on FDI, a sort of investment that generates international capital flows [2]. As per Kamran, Chaudhry [3], FDI has expanded faster than

**Competing interests:** The authors have declared that no competing interests exist.

most other forms of global trade. The analysis of logistic performance, global competitiveness, and IR as FDI flow-influencing factors are given priority in this study. In doing so, it fills the void since many researchers have not given sufficient attention to identify the impact of these factors on FDI in the African region.

Logistics plays a crucial part in easing commerce, decreasing transportation costs, and boosting overall economic development [2, 4, 5]. In 2007, the World Bank announced the Logistic Performance Index (LPI). The LPI, commonly referred to as the "Connecting to Compete" report, offers the most comprehensive international comparison tool for a country's ability to facilitate trade and transportation [6]. Siddiqui and Vita [7] reported that knowledge of countries' elements can increase their freight transport efficiency and uncover problems by monitoring their trade and logistics performance. The international LPI shows how well a country performs in logistics, but it cannot accurately estimate the associated costs. LPI is further subdivided into the following categories: Customs, Infrastructure, Ease of Shipment, Logistics Services, Ease of Tracking, Timeliness and Domestic Logistics Costs. Martí, Puertas [8] observed that LPI had promoted trade and transportation while evaluating customs procedures, logistics costs, and quality of transportation infrastructure between countries.

The Global Competitiveness Index (GCI) was established in 2004 by the World Economic Forum to rank the competitiveness of countries worldwide [9]. GCI examines both microeconomic and macroeconomic elements that have significant impact on the economic potential of a nation [10]. Based on statistical information gathered from internationally recognised organizations such as the International Monetary Fund, World Health Organization, USA-India Educational Foundation, and others, GCI compares countries' economic competitiveness. It consists of 12 pillars: institutions, infrastructure, macroeconomic stability, health, skills, product market, labour market, financial system, ICT adoption, market size, business dynamism, and innovation capability. GCI assigns larger relative weights for those relatively more important pillars for an economy, given its specific level of development considering into account its developmental stages. Any country that lies halfway between two stages is considered to be in the transition period [11]. GCI assesses countries both in the long and short term from corporate and economic standpoints [12].

IR, the cost of borrowing, is determined by monetary supply and demand [13, 14]. Here, IR can be simply defined as the return on investment earned by the investor. Capital debtors must pay IR in repayment of loans, or a sum of money borrowed from a capital lender. The economic studies highlight various types of IR, such as nominal IR, lending IR, real IR (RIR), etc. However, depending on the type of IR, the effect it causes on the investors differ. Alongside the African regional RIR fluctuations, the latter is one of the regions with the highest RIR globally. Considering the availability of data RIR, which is the lending IR adjusted from the inflation factor, will be utilised for the analysis of this study.

The objective of this study is to investigate the impact of LPI, GCI, and IR on FDI in the African region. It was determined that Africa had been the lowest FDI attractor during the last few decades. Therefore, this study is an utmost necessity to understand how the competitiveness of the region, and infrastructural availability have contributed to attracting FDI. This study differs from past studies in three ways. Firstly, it fills a knowledge gap that exists due to a lack of studies in the African region to examine the combined impact of the LPI, GCI, and IR variables on FDI. Secondly, this study investigates the effect of LPI, GCI, and IR on FDI in the countries in the African region by conducting a country specific analysis. Lastly, this study provides a thorough comparison and deeper understanding of the impacts and the differences in the impacts of the control variables on FDI between countries in the African region.

The remaining sections of the study are structured as follows. The Literature Review consist of findings of the past studies, the Data and Methodology presents the sources of gathered data

and the employed analytical techniques. The Results and Discussion section provides a holistic analysis of the study's findings and, finally, the study concludes with Conclusion and Recommendations.

## Literature review

According to past studies, various factors influence FDI inflows into the economies and impacts that these factors have on FDI inflows vary between regions and countries globally due to several factors. The past literature was reviewed in this current study focusing on the African region to analyse the past findings on the impact of LPI, GCI and IR on FDI inflows.

Africa has become an attractive FDI destination due to the ratification of the 'African Continental Free Trade Area', a pact which promotes trade and investment within the region. Many studies have been conducted to identify the factors that impact FDI. Some studies, among many factors, highlighted that LPI has a significant positive impact on FDI, and the logistics performance of an economy can be considered an important factor affecting FDI inflows [15–17]. Similarly, a study identified that FDI positively correlates with green logistics performance in the Belt and Road Initiative countries [18]. However, in contrast, Zaman [19] determined a significant negative relationship between the environmental impact indicator of LPI and FDI inflows in the Brazil, Russia, India, China and South Africa (BRICS) countries.

Having lower GCI ranking countries despite the fact that Africa is a desirable location for FDI has a significant disadvantage. Moreover, it was identified that GCI has a significant positive impact on FDI inflows. The former can be considered a determinant of FDI and an important benchmark for investors [20–22]. Similarly, Lysandrou, Solomon [23] confirmed that the quality of public governance institutions of a country, measured by the GCI, is significantly and positively correlated with FDI inflows. Marby and Chen [24] deployed GCI to explore how to construct a dynamic and aggregated index of FDI factors and deterrents to assist businesses with their FDI plans. It is because the level of competitiveness has an influence on FDI inflows and accounts for risks that are not reflected in real GDP. Therefore, according to past studies GCI positively influences FDI.

Various studies have analysed the influence of IR on FDI in the African region. Two studies were conducted to investigate the impact of IR on FDI in Ghana. Here, scholars emphasised countries tend to attracting more FDI when the IR value is decreasing, showing IR has a substantial negative influence on FDI. However, authors further suggested that the characteristics of countries indicated a substantial influence on the nature of the IR impact on FDI as well [25, 26]. According to Musyoka [27] and Nyanyuki [28], both RIR and currency rates negatively and significantly influence FDI inflow in Kenya. In contrast, a study in Kenya highlighted that the IR has a favourable impact on FDI inflows to Kenya's energy and petroleum sector [29]. Another study by Ezeoha and Cattaneo [30] emphasised that in Sub-Saharan Africa, the RIR indicated a significant negative impact on FDI and proving that RIR stability is a key macroeconomic factor that draws FDI. Faroh and Shen [31] show the past 20 years of African have seen larger IR volatility. FDI inflows in Sierra Leone were significantly influenced by other macroeconomic factors, however there was no conclusive evidence between IR and FDI inflows.

Only a limited number of studies have been conducted to analyse the impact of LPI, GCI and IR individually on FDI. According to information available to the authors of the current study, within the African region, no studies have been conducted to analyse the impact of these control variables on FDI for the region as a whole or a country wise analysis. Therefore, this study aims to fill this research gap by analysing the impact of LPI, GCI and RIR on FDI in the African region and individual countries within the region.

## Data and methodology

This study was reviewed and approved by Sri Lanka Institute of Information Technology (SLIIT) Business School and the SLIIT ethical review board. Study used the secondary data sources and the data file used for the study is presented in S1 Appendix. The study used a panel dataset of 26 countries, with an emphasis on the African continent, during a 6-year period (2007–2018), including the years 2007, 2010, 2012, 2014, 2016, and 2018. Availability of secondary data was taken into consideration when choosing the sample. Due to the limited availability of data on LPI (only for the six years stated above), there are gaps in the time series. In addition, the Global Competitiveness Report released by the World Economic Forum for GCI and the World Bank's World Development Indicators for LPI, FDI, and IR were secondary data sources. To analyse how LPI, GCI, and IR affect FDI with a focus on the African region, Eq 1 below was developed while Eq 2 was developed for analysing how LPI, GCI, and IR affect FDI in different countries within the region.

$$\ln FDI_{it} = \beta_0 + \beta_1 LPI_{it} + \beta_2 GCI_{it} + \beta_3 IR_{it} + \varepsilon_{it} \tag{1}$$

$$\ln FDI_t = \beta_0 + \beta_1 LPI_t + \beta_2 GCI_t + \beta_3 IR_t + \varepsilon_t \tag{2}$$

The $\ln FDI_{it}$ denotes the natural log value of the FDI inflow measured by the current USD. The natural log values of FDI are utilised in order to ensure normal distribution of data. $LPI_{it}$ denotes the overall LPI value, $GCI_{it}$ denotes the overall GCI value and $IR_{it}$ denotes the RIR. The i denotes the country and t denotes the time period. The $\beta_x$ represents the coefficients of variables and $\varepsilon_{it}$ denotes the error term of the regression equation.

Panel regression and Multiple Linear Regression (MLR) models were used as the analytical techniques. Two specification tests were undertaken to assess the appropriate model namely, Random Effect (RE), Fixed Effect (FE) or Pooled Ordinary Least Square model. The Hausman test was used to evaluate the applicability of RE and FE models, while the Breusch Pagan test was used to choose the best suited model from the Pooled Ordinary Least Square and RE models. Furthermore, the analysis was conducted using standard robust error eliminating the impact of the heteroscedastic issue. However, past studies further justify that when the sample size is larger the impact of the normality and the collinearity issues are significantly minimised [32–34]. Moreover, scatter plot diagrams with linear fit of LPI, GCI and IR were employed to illustrate the country specific trends in the impacts of the control variables on FDI in countries in the African region.

## Results

An overview of the research findings is provided in this section. The Stata, a statistical software, was used for the analysis of data. The descriptive statistics of the variables for the chosen time period are shown in S2 Appendix. Using 113 observations, descriptive statistics for the African region were analysed. The average FDI, GCI, LPI, and IR values are 1.01 USD billion, 3.56, 2.53, and 7.38, respectively. South Africa holds the highest average for FDI, while Angola remains the lowest. Mauritius has the highest GCI average, while Angola continues to have the lowest. While the LPI average for Sierra Leone is the lowest, it is the greatest in South Africa. The average for IR is now the highest in the Gambia and the lowest in Zimbabwe.

To produce more robust estimates, whether to utilise the regressions of the pooled ordinary least squares (POLS), random-effect (RE), and fixed-effect (FE) are verified in detail. For the selection of estimation approaches of the POLS and RE regression, a Breusch Pagan Test was employed. The result displays that $\text{Chi}^2$ (3) = 80.00 with a $p$-value $< 0.01$, indicating that the PLOS is inappropriate. Simultaneously, Hausman test was utilised, and the result suggests that

Chi$^2$ (3) = 2.99 with a $p$-value > 0.10, indicating that the FE regression is not appropriate. Thus, this study utilises the approach of the RE regression to evaluate the impacts of LPI, GCI, and IR on FDI in the African Region.

Further, MLR models were employed to examine the impact of the control variables on FDI in selected countries within the region. The results of the RE and MLR models employed to investigate the impact of LPI, GCI and IR on FDI are portrayed in S3 Appendix, including the standardised coefficients and the error components for the African region and selected countries within the region. S4 Appendix illustrates the scatter plot diagrams with a linear fit for the countries within the African region.

In the African region, LPI indicates a significant positive impact on the FDI flows at a 10% significance level. Similarly, LPI indicates a significant positive impact on the FDI at 1%, 5% and 10% significance levels in Gambia, Lesotho and Rwanda, respectively. However, in contrast to the majority of the findings, LPI indicates a significant negative impact on FDI at a 10% significance level in Mauritius islands.

When considering the impact of GCI on FDI in Algeria and Lesotho, there is a significant positive impact at 5% and 1% levels of significance, respectively, similar to the findings of Marby and Chen [24]. In contrast, GCI indicates a significant negative impact on FDI in Rwanda at a 1% significance level and in Mauritius and Namibia at a 10% significance level.

IR indicates a significant negative impact on FDI flows at a 5% significance level in the African region which is a similar result obtained compared to the existing literature [26–28]. Similarly, IR has a significant negative impact on FDI at a 1% significance level in Algeria and Rwanda and at a 5% significance level in Mauritius. However, in Lesotho, IR indicates a significant positive impact on FDI at a 1% level of significance and a similar result was obtained by [29].

## Discussion

This study identified that LPI, GCI, and IR are key factors that have an impact on the FDI flows in the African region and individual countries within the region. According to the results, only the LPI and IR significantly impact FDI in the African region. However, according to the country specific results, GCI also significantly impacts FDI for certain countries. Apart from these, variations are evident about how these variables impact FDI between different countries, where the impact is positive in some countries while negative in others. It can be concluded that the control variables do not individually impact FDI flow, but various other factors have contributed in this sense.

Moreover, focusing on the bigger picture, the African region can be identified as the lowest FDI attracting destination compared to the other regions and the difference between the FDI flow is also significant. As mentioned earlier, the reason for such a lag behind could also be the effect of other regional factors.

Factors such as institutional quality, trade and transport related infrastructure, customs clearance efficiency, competency of logistics services etc., have improved logistics performance in the African region [18]. Consequently, these have contributed to LPI positively impacting on FDI. Similarly, factors such as institutional quality can be considered as a factor influence the effect of GCI on FDI inflow [23]. Moreover, the sub pillars of GCI separately impact on FDI. The competitiveness indices such as market size and the subcomponents of higher education and training are key components improving FDI inflows in the country [22]. This implies the positive and negative impacts of different countries and GCI for not having a significant impact on FDI in the African region as a whole. The impact of IR on FDI differs with the types of IR and investments. Investors undertaking investments prefer higher IR to gain a higher

return, while investors seeking financing prefer lower lending IR to reduce their cost of capital.

Further consideration on IR, the African region has the highest IR values among other regions. It can be argued that the regional investment strategy was to maintain higher IR which influence investors to invest more to gain high return on investments. However, strategy may have not provided the expected outcomes because the investors not only focus the return on investment but also the other factors, such as risk and growth potential.

Terrorism has been a major drawback for African region countries during past two decades in all types of economic activities. The known active militant Islamist terrorist groups including the ISIS, Al Qaeda, Boko Haram and Al Shabaab have been conducting attacks throughout the region while mainly operating in Chad, Algeria, Mauritania, Mali and Niger soil. As per the UNCTAD [35], Egypt, Nigeria, Congo and South Africa, the countries mostly with low terrorist activities, have managed to attract FDI over United States Dollars (USD) 3 billion (Bn). Therefore, the impact of LPI, GCI and IR on FDI flow varies mostly on the country specific characteristics of the African regional countries.

## Conclusion

Despite LPI and GCI being two key global indexes and IR being a major macroeconomic indicator, only limited studies had been conducted to analyse the impact of LPI, GCI and IR on FDI in countries within the African region. Therefore, this study contributes significantly by analysing the impact of LPI, GCI and IR on FDI as a whole in the African region and individual countries within the region. This study was conducted by utilising data from 26 countries in the African region for the six years including 2007, 2010, 2012, 2014, 2016 and 2018, and models of Panel Regression and MLR were employed data analysis to determine the regional and country specific impacts respectively.

The study findings indicate mixed results justifying that LPI has a significant positive impact on FDI similar to many past studies [15–17]. This means that LPI is a global indicator of location choice of investors while IR significantly negatively impacts FDI inflows in the African region aligning with past studies [27, 28, 30]. This indicates that higher IR discourages investors and IR stability encourages FDI. Moreover, the country specific results concluded that GCI too, had a significant effect on FDI inflows in the African region, which is a finding backed by many past studies [20–22]. This indicates that all the three control variables have a significant impact on FDI in the African region. However, the African region consists of underdeveloped countries with high poverty levels, political crises and terrorism that discourage foreign investors due to high unstable economic conditions [36]. As a result, it has led the African region to lower rankings in all the indexes and attract a considerably low amount of FDI.

### Policy implications

This study contributes to understanding the policies that need to be imposed to attract FDI inflows into an economy. Given the vast challenges African countries face due to poor economic status, they must revisit policy planning mechanisms and strengthen the macroeconomic environment. As key actors in developing nations, policymakers should plan and construct contemporary transportation and logistics, interconnected technology components as well as infrastructural development, that can be employed for long-term economic growth will attract FDI [17]. Generally, when the cost of capital in an economy is lower, it can attract more FDI into the economies. Therefore, the RIR in an economy should be maintained at lower levels. Furthermore, higher levels of competitiveness will attract more FDI and hence,

policymakers should focus on maintaining appropriate levels of competitiveness. Additionally, Africa as a region with high uncertainties and crises, need to implement favourable policies that can provide a conducive setting where investors perceive the country as low risk, low uncertainty etc., which can be more advantageous to attract FDI in the long run.

The limitations of this study include omitting a number of countries due to lack of data availability. In addition, this study has not focused on a pillar-wise analysis for LPI and GCI to understand how each key indicator impacts FDI flows, indicating the inability to capture some useful discoveries in this setting.

A global analysis can be conducted for future research to determine the impact of LPI, GCI and IR on FDI in all continents and other factors affecting FDI. Moreover, a broader study can be carried out by analysing the impacts of each of the LPI and GCI sub-pillars on FDI.

## Supporting information

**S1 Appendix. Data file.**
(XLSX)

**S2 Appendix. Summary of descriptive statistics of variables.**
(DOCX)

**S3 Appendix. RE and MLR model results.**
(DOCX)

**S4 Appendix. Linear fit scatter plot graphs.**
(DOCX)

## Acknowledgments

The authors would like to thank Ms. Gayendri Karunarathne for proof-reading and editing this manuscript.

## Author Contributions

**Conceptualization:** Sashini Rathnayake, Sanjula Jayakody, Ruwan Jayathilaka.

**Data curation:** Sashini Rathnayake, Sanjula Jayakody, Pasindu Wannisinghe, Deshani Wijayasinghe.

**Formal analysis:** Sashini Rathnayake, Sanjula Jayakody, Pasindu Wannisinghe, Deshani Wijayasinghe.

**Investigation:** Sashini Rathnayake, Sanjula Jayakody, Pasindu Wannisinghe.

**Methodology:** Sashini Rathnayake, Sanjula Jayakody, Pasindu Wannisinghe, Deshani Wijaya-singhe, Ruwan Jayathilaka.

**Project administration:** Ruwan Jayathilaka, Naduni Madhavika.

**Resources:** Naduni Madhavika.

**Software:** Sashini Rathnayake, Sanjula Jayakody, Pasindu Wannisinghe.

**Supervision:** Ruwan Jayathilaka, Naduni Madhavika.

**Validation:** Ruwan Jayathilaka, Naduni Madhavika.

**Visualization:** Sashini Rathnayake, Sanjula Jayakody, Pasindu Wannisinghe, Deshani Wijayasinghe.

**Writing – original draft:** Sashini Rathnayake, Sanjula Jayakody, Pasindu Wannisinghe, Ruwan Jayathilaka, Naduni Madhavika.

**Writing – review & editing:** Ruwan Jayathilaka.

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
