## [Decision Letter · Decision Letter 0]

8 Dec 2022

PONE-D-22-25104LPI, GCI and IR as Determinants of FDI in the African RegionPLOS ONE

Dear Dr. Jayathilaka,

Thank you for submitting your manuscript to PLOS ONE. After careful consideration, we feel that it has merit but does not fully meet PLOS ONE’s publication criteria as it currently stands. Therefore, we invite you to submit a revised version of the manuscript that addresses the points raised during the review process.

We look forward to receiving your revised manuscript.

Kind regards,

Wajid Khan

Academic Editor

PLOS ONE

Journal Requirements:

Reviewers' comments:

Reviewer's Responses to Questions

**Comments to the Author**

1. Is the manuscript technically sound, and do the data support the conclusions?

Reviewer #1: Yes

Reviewer #2: Yes

2. Has the statistical analysis been performed appropriately and rigorously? 

Reviewer #1: Yes

Reviewer #2: No

3. Have the authors made all data underlying the findings in their manuscript fully available?

Reviewer #1: Yes

Reviewer #2: Yes

4. Is the manuscript presented in an intelligible fashion and written in standard English?

Reviewer #1: Yes

Reviewer #2: Yes

5. Review Comments to the Author

Reviewer #1: The paper is relevant for the journal. However, in order to be published, it needs some innovation in structuring. For example, the paper should have more weight in the title and in the abstract, and the innovative methodology should be emphasized.. Please adjust the result, conclusions also accordingly. The method in itself is good and well described.

Reviewer #2: On the basis of my observations and extensive check, I have the following minor suggestions that should be addressed:

1. The title of the study is simple, it should be changed to more suitable.

2. The background/introduction of the study has a lack of new studies to be cited.

3. The methodology section is week, where is the model specification for this study to address the variables? there should be a model specification for this paper.

4. The estimated model(s) lack proper diagnostic checking and stationarity tests should be included.

5. The estimated model(s) may be strengthened by applying co integrating techniques such as panel ARDL, if possible.

6. The conclusion section is totally week, rewrite it again according to the study findings.

6. PLOS authors have the option to publish the peer review history of their article (what does this mean?). If published, this will include your full peer review and any attached files.

Reviewer #1: No

Reviewer #2: No

---

## [Author Response · Author response to Decision Letter 0]

26 Dec 2022

Point by point response to reviewers

Dear editor and reviewers.

Thank you for giving us the opportunity to submit a revised draft of the manuscript “LPI, GCI and IR as Determinants of FDI in the African Region” for publication in the prestigious “PLOS ONE” journal. As per the reviewers’ suggestion title has changed to “Macroeconomic Factors Affecting FDI in the African Region”. We appreciate the time and effort that you have dedicated to providing feedback on our manuscript and are grateful for the insightful comments on and valuable improvements to our paper. We have incorporated most of the suggestions made by the editor and respective reviewers. Those changes are highlighted within the manuscript. Please see below, for a point-by-point response to the reviewers’ comments and concerns. All page numbers refer to the revised manuscript file with tracked changes.

Reviewer 1 comment: The paper is relevant for the journal. However, in order to be published, it needs some innovation in structuring. For example, the paper should have more weight in the title and in the abstract, and the innovative methodology should be emphasized.. Please adjust the result, conclusions also accordingly. The method in itself is good and well described.

Authors’ Response: Well noted and thank you. Based on the improvements pointed out, the manuscript was revised as below.

Title and Abstract were improved and changed accordingly. New title is “Macroeconomic Factors Affecting FDI in the African Region”. Refer line 1 to 2 and 29 to 30 for mentioned section. Refer Reviewer #2 Comment 1.

“Foreign Direct Investment (FDI) occurs when one country invests in another. Multiple factors have contributed to fluctuations in FDI flows globally. This study investigates the impact of the Logistics Performance Index (LPI), Global Competitiveness Index (GCI) and Interest Rates (IR) on FDI in the African region. The study is significant because the African region is underdeveloped and with an unstable macroeconomic environment. Data were collected for 26 countries in the African region for the years 2007, 2010, 2012, 2014, 2016 and 2018 and analysed using Panel Regression and Multiple Linear Regression models. The study's findings concluded that LPI, GCI, and IR are three major macroeconomic factors impacting FDI inflows. The results indicated that LPI positively impacts FDI in Gambia, Lesotho and Rwanda, while in contrast, LPI impacts FDI negatively in Mauritius. GCI has a positive impact on FDI in Algeria and Lesotho with a negative impact in Rwanda, Mauritius and Namibia. Moreover, IR has a negative impact on FDI in Algeria, Rwanda and Mauritius with a positive impact in Lesotho. Policymakers should pay more attention to the infrastructure development and management of macroeconomic and other factors affecting FDI.” Refer line 33 to 47 for mentioned section.

The methodology section was improved by emphasizing the innovative methodology with further explanations on the selection and validity of the methodology. The results section was further improved accordingly. Further, refer Reviewer #2 Comments 3, 4 and 5.

The conclusion section was adjusted and rewritten accordingly. Refer Reviewer #2 Comment 6.

Reviewer 2 comment 1: The title of the study is simple, it should be changed to more suitable.

Authors’ Response: Title has been changed to “Macroeconomic Factors Affecting FDI in the African Region”. Refer line 1 to 2 and 29 to 30 for mentioned section.

Reviewer 2 comment 2: The background/introduction of the study has a lack of new studies to be cited.

Authors’ Response: Thank you for the comment and it was well received. The new studies have been incorporated in the manuscript as follows.

“…… A country's economic development is primarily reliant on FDI, a sort of investment that generates international capital flows [2].” Refer line 61 to 62 for mentioned section.

 “Logistics plays a crucial part in easing commerce, decreasing transportation costs, and boosting overall economic development [2, 4, 5].” Refer line 68 to 69 for mentioned section.

“……GCI examines both microeconomic and macroeconomic elements that have significant impact on the economic potential of a nation [10].” Refer line 81 to 83 for mentioned section.

“IR, the cost of borrowing, is determined by monetary supply and demand [13, 14].” Refer line 93 for mentioned section.

Reviewer 2 comment 3: The methodology section is week, where is the model specification for this study to address the variables? there should be a model specification for this paper.

Authors’ Response: Thank you for the comment and it is well noted. As the model specification tests Breusch Pagan test and the Hausman test

were conducted to determine the most suitable panel regression model out of POLS model, Fixed Effect Model and the Random Effect Model. To further highlighted the model specification test we have included the results of the test in the results section in a table and it is further mentioned in the text as well.

“To produce more robust estimates, whether to utilise the regressions of the pooled ordinary least squares (POLS), random-effect (RE), and fixed-effect (FE) are verified in detail. For the selection of estimation approaches of the POLS and RE regression, a Breusch Pagan Test was employed. The result displays that Chi2 (3) = 80.00 with a p-value < 0.01, indicating that the PLOS is inappropriate. Simultaneously, Hausman test was utilised, and the result suggests that Chi2 (3) = 2.99 with a p-value > 0.10, indicating that the FE regression is not appropriate. Thus, this study utilises the approach of the RE regression to evaluate the impacts of LPI, GCI, and IR on FDI in the African Region.” Refer line 206 to 213 for mentioned section.

Reviewer 2 comment 4: The estimated model(s) lack proper diagnostic checking and stationarity tests should be included.

Authors’ Response: Thank you for pointing out this. In this study we have utilized the standard robust error which leads to eliminate the effect of heteroscedasticity issue. However, this study was conducted African region utilizing more than 50% of the African region countries based on the availability of the secondary data. According to econometrics text books and past studies, when the sample size is large the normality and collinearity issue become less impactful on the results of the econometric technique utilized. 

 “Furthermore, the analysis was conducted using standard robust error eliminating the impact of the heteroscedastic issue. However, past studies further justify that when the sample size is larger the impact of the normality and the collinearity issues are significantly minimised [32-34].” Refer line 188 to 191 for further justification on this.

When it comes to the stationary test, in this study even though the panel dataset consists of 33 cross sections it contains only 6 time series which are 2007, 2010, 2012, 2014, 2016, and 2018 with uneven gaps. Therefore, the dataset is unbalanced with uneven gaps in the time series. Therefore, unfortunately we were unable to conduct the stationarity test as a result of the constraint arose due to the limitation of the dataset.

Reviewer 2 comment 5: The estimated model(s) may be strengthened by applying co integrating techniques such as panel ARDL, if possible.

Authors’ Response: Thank you for the comment. We understand that conducting a ARDL model would further strengthen this study. However, since there are uneven gaps in the panel dataset as it was mentioned in the Reviewer 02 Comment 04. Therefore, conducting an ARDL model for this study was not possible option.

Reviewer 2 comment 6: The conclusion section is totally week, rewrite it again according to the study findings.

Authors’ Response: Thank you for the comment and it was well received. The conclusion more strengthened and rewritten by separating the policy implications. 

“ Conclusion

Despite LPI and GCI being two key global indexes and IR being a major macroeconomic indicator, only limited studies had been conducted to analyse the impact of LPI, GCI and IR on FDI in countries within the African region. Therefore, this study contributes significantly by analysing the impact of LPI, GCI and IR on FDI as a whole in the African region and individual countries within the region. This study was conducted by utilising data from 26 countries in the African region for the six years including 2007, 2010, 2012, 2014, 2016 and 2018, and models of Panel Regression and MLR were employed data analysis to determine the regional and country specific impacts respectively.

The study findings indicate mixed results justifying that LPI has a significant positive impact on FDI similar to many past studies [15-17]. This means that LPI is a global indicator of location choice of investors while IR significantly negatively impacts FDI inflows in the African region aligning with past studies [27, 28, 30]. This indicates that higher IR discourages investors and IR stability encourages FDI. Moreover, the country specific results concluded that GCI too, had a significant effect on FDI inflows in the African region, which is a finding backed by many past studies [20-22]. This indicates that all the three control variables have a significant impact on FDI in the African region. However, the African region consists of underdeveloped countries with high poverty levels, political crises and terrorism that discourage foreign investors due to high unstable economic conditions [36]. As a result, it has led the African region to lower rankings in all the indexes and attract a considerably low amount of FDI.”

Refer lines 282 to 303 for mentioned section.

“Policy Implications

This study contributes to understanding the policies that need to be imposed to attract FDI inflows into an economy. Given the vast challenges African countries face due to poor economic status, they must revisit policy planning mechanisms and strengthen the macroeconomic environment. As key actors in developing nations, policymakers should plan and construct contemporary transportation and logistics, interconnected technology components as well as infrastructural development, that can be employed for long-term economic growth will attract FDI [17]. Generally, when the cost of capital in an economy is lower, it can attract more FDI into the economies. Therefore, the RIR in an economy should be maintained at lower levels. Furthermore, higher levels of competitiveness will attract more FDI and hence, policymakers should focus on maintaining appropriate levels of competitiveness. Additionally, Africa as a region with high uncertainties and crises, need to implement favourable policies that can provide a conducive setting where investors perceive the country as low risk, low uncertainty etc., which can be more advantageous to attract FDI in the long run.” Refer lines 304 to 318 for mentioned section.

---

## [Decision Letter · Decision Letter 1]

10 Jan 2023

Macroeconomic Factors Affecting FDI in the African Region

PONE-D-22-25104R1

Dear Dr Ruwan Jayathilaka,

We’re pleased to inform you that your manuscript has been judged scientifically suitable for publication and will be formally accepted for publication once it meets all outstanding technical requirements.

Kind regards,

Wajid Khan

Academic Editor

PLOS ONE

Additional Editor Comments (optional):

Reviewers' comments:

Reviewer's Responses to Questions

**Comments to the Author**

1. If the authors have adequately addressed your comments raised in a previous round of review and you feel that this manuscript is now acceptable for publication, you may indicate that here to bypass the “Comments to the Author” section, enter your conflict of interest statement in the “Confidential to Editor” section, and submit your "Accept" recommendation.

Reviewer #2: All comments have been addressed

2. Is the manuscript technically sound, and do the data support the conclusions?

Reviewer #2: Yes

3. Has the statistical analysis been performed appropriately and rigorously? 

Reviewer #2: Yes

4. Have the authors made all data underlying the findings in their manuscript fully available?

Reviewer #2: Yes

5. Is the manuscript presented in an intelligible fashion and written in standard English?

Reviewer #2: Yes

6. Review Comments to the Author

Reviewer #2: All the questions raised in the review were answered satisfactorily. The weaknesses and errors have also been improved/corrected.

7. PLOS authors have the option to publish the peer review history of their article (what does this mean?). If published, this will include your full peer review and any attached files.

Reviewer #2: No

---

## [Editor Report · Acceptance letter]

12 Jan 2023

PONE-D-22-25104R1 

Macroeconomic Factors Affecting FDI in the African Region 

Dear Dr. Jayathilaka:

I'm pleased to inform you that your manuscript has been deemed suitable for publication in PLOS ONE. Congratulations! Your manuscript is now with our production department. 

Kind regards, 

on behalf of

Dr. Wajid Khan 

Academic Editor

PLOS ONE